# Impact of blood meals taken on ivermectin-treated livestock on survival and egg production of the malaria vector *Anopheles coluzzii* under laboratory conditions

**Sié Hermann Pooda**[1,2]*, **Domonbabele François de Salle Hien**[3], **Soumaïla Pagabeleguem**[1,2], **Andre Patrick Heinrich**[4], **Angélique Porciani**[5], **André Barembaye Sagna**[5], **Lamidi Zela**[6], **Lassane Percoma**[2], **Thierry Lefèvre**[5], **Roch Kounbobr Dabiré**[3], **Amnan Alphonsine Koffi**[7], **Rolf-Alexander Düring**[4], **Cédric Pennetier**[5], **Nicolas Moiroux**[5], **Karine Mouline**[5]*

1 Université de Dédougou, Dédougou (UDDG), Dédougou, Burkina Faso, 2 Insectarium de Bobo-Dioulasso —Campagne d'Eradication de la Mouche Tsé-tsé et de la Tryapnosomose (IBD-CETT), Bobo-Dioulasso, Burkina Faso, 3 Institut de Recherche en Sciences de la Santé (IRSS), Bobo-Dioulasso, Burkina Faso, 4 Institute of Soil Science and Soil Conservation, Research Center for Biosystems, Land Use and Nutrition (iFZ), Justus Liebig University Giessen, Giessen, Germany, 5 Unité Mixte sur les Maladies Infectieuses et Vecteurs: Ecologie, Génétique, Evolution et Contrôle (MIVEGEC), Université de Montpellier, IRD, CNRS, Montpellier, France, 6 Centre International de Recherche-Développement sur l'Elevage en zone Subhumide (CIRDES), Bobo-Dioulasso, Burkina Faso, 7 Institut Pierre Richet/Institut National de Santé Publique, Bouaké, Côte d'Ivoire

* poodaisehermann@yahoo.fr (SHP); karine.mouline@ird.fr (KM)

**Data Availability Statement:** All data supporting these results are available in the Supporting Information S3 File to S5 File.

## Abstract

Treatment of livestock with endectocides such as ivermectin is viewed as a complementary vector control approach to address residual transmission of malaria. However, efficacy of this treatment may vary between animal species. Hence, our purpose was to investigate the effects of ivermectin treatments of common livestock species on life history traits of the opportunistic malaria vector *Anopheles coluzzii*. Sheep, goats and pigs were treated using injectable veterinary ivermectin formulation at the species-specific doses (recommended dose for all species and high dose in pig). Mosquito batches were exposed to treated and control (not injected) animals at different days after treatment. Daily mosquito mortality was recorded and fecundity assessed through the count of gravid females and the number of eggs they developed. The recommended dose of ivermectin induced a significant decrease in mosquito survival for up to 7 days after injection (DAI), with a decrease of 89.7%, 66.7%, and 48.4% in treated pigs, goats and sheep, respectively, compared to control animals. In treated pigs, the triple therapeutic dose decreased mosquito survival of 68.97% relatively to controls up to 14 DAI. The average number in gravid females *Anopheles* that survived after feeding on treated animals were reduced when blood-meals were taken on sheep (2.57% and 42.03% at 2 and 7 DAI), or on goats (decrease of the 28.28% and 73.64% respectively at 2 and 7 DAI). This study shows that ivermectin treatments to animals negatively impacts *An. coluzzii* life history traits and could reduce vector densities in areas where livestock live near humans. However, due to short-term efficacy of single dose treatments, repeated treatments and potentially increased dosages would be required to span the transmission

**Funding:** This experimental study was supported by the REACT project founded by the "Expertise France" and the LAMIVECT (LAboratoire Mixte International sur les Maladies à VECTeurs) founded by the Institut de Recherche pour le Développement (IRD).

**Competing interests:** The authors have declared that no competing interests exist.

season. The use of long-acting ivermectin formulations is discussed as a mean for extending efficacy while remaining cost effective.

## Introduction

A remarkable reduction in the burden of malaria has been achieved globally over the past two decades. The incidence of the disease has declined from 82 per 1000 population at risk in 2000 to 57 in 2019. The malaria mortality rate has also decreased over the same period, from 30 per 100 000 population at risk to 15 per 100 000 in 2019 [1]. However, an increasing trend in malaria cases has been observed globally over the past 5 years. Malaria cases have increased from 218 million to 247 million between 2019 and 2021 [2], indicating that the current malaria prevention strategies are no longer adequate to control the disease and that new and/or complementary measures are urgently needed.

Indoor vector control tools such as LLINs and IRS have been critical in decreasing the burden of malaria by targeting anthropophilic (prefer to feed on humans), endophagic (prefer to bite indoors), and endophilic (prefer to rest indoors) malaria vectors [3]. Their wide-scale deployment has contributed to avoid approximatively 1.5 billion malaria cases between 2000 and 2019 [2]. However, these interventions have had limited impact on exophagic (prefer to bite outdoors), exophilic (prefer to rest outdoors) and zoophagic/opportunistic (blood feed on animals/do so occasionally when human are not available) malaria vectors such as *An. arabiensis* and *An. coluzzii* [4]. This could explain in part the high malaria transmission observed in many African countries where malaria vectors exhibit such characteristics, despite high coverage of LLINs and IRS [5].

*Anopheles coluzzii*, formerly *An. gambiae* M form, plays a significant role in malaria transmission in west Africa [6]. However, this highly anthropophilic species is increasingly reported to feed on non-human hosts (*i.e.* cattle, goats, dogs) [7–9]. This plastic feeding strategy of this yet highly innate anthropophagic species is thought to be the consequence of reduced accessibility of human hosts due to widescale deployment of LLINs and IRS, and the accessibility of close and readily accessible domestic animals hosts in agropastoral settings [7–9].

The main challenges with LLINs and IRS strategies are the persistence of residual transmission of *Plasmodium* due to mosquito populations resisting to insecticides (*i.e.* those that carry metabolic and target site mutations), or displaying behaviors that limit or avoid the contact with the molecules (*i.e.* Those that feed on livestock, bite and resting outdoors, display early or late aggressive behavior, exit early from houses to evade indoor insecticide exposure, etc.). Human host behaviors may also favor residual transmission through exposure to vectors (*i.e.* by staying or sleeping indoor or outdoor without protective measures). To target endophilic insecticide-resistant malaria vectors, new generations of LLINs (*i.e.* dual-active ingredients LLINs) were developed [10,11] and are now widely distributed to replace standard nets [12]. These tools will need, however, to be combined with complementary interventions that can target all vectors that are yet out of reach to achieve the malaria elimination goal. Hence, several interventions are currently under investigations, such as treating livestock with endectocides (*i.e.* ivermectin) [13], improving housing [14], the use of attractive toxic sugar baits [15], and larval source management [16].

In this context, studies have documented the efficacy of ivermectin in killing mosquitoes that took a blood meal on treated animals. The mosquitocidal effect has been demonstrated in major malaria vectors on cattle [17–19], pigs [20] and dogs [21]. Small ruminants such as

sheep and goats are not yet considered for this approach. These animals that usually live near human populations also represent an alternative blood source for malaria vectors that enables their reproduction and survival, hence sustaining their role in *Plasmodium* transmission and leaning toward zoopotentiation [22,23]. Therefore, using ivermectin for treating a large panel of peridomestic animals would represent an endectocide-based zooprophylactic approach, in the frame of the One-Health concept, which would virtuously intricate human's and animal's health [24].

Burkina Faso has shown similar malaria epidemiological trend than worldwide, owing to the intensive use of core interventions recommended by the World Health Organization (WHO) to fight the disease: chemoprevention (for pregnant women and children under 5 years), treatment of diagnosed cases with artemisinin-based combination therapies, and vector control (long-acting insecticide-treated mosquito nets, (LLINs), and indoor residual spraying (IRS) [25,26]. However, in areas where human live in close vicinity to domestic animals, which represent the largest, Soudano-Sahelian, part of the country, major malaria mosquitoes are zoophagic [7,27]); hence, targeting the animal component of malaria transmission using complementary approaches like endectocide treated livestock is there essential. According to Imbahale et al. [28], Burkina Faso is among the countries where such approach should be efficient at reducing malaria incidence.

The pharmacokinetics of ivermectin varies according to animal species, injected dose, and route of administration [29,30]. The systemic insecticidal effects of the drug on mosquitoes may then vary depending on the treated animal species, as it may also depend on the *Anopheles* vector species considered [31,32]. In the present study, a laboratory experiment was performed in Burkina Faso to investigate the effects of ivermectin on the survival and egg production of field derived *An. coluzzii* fed on treated local sheep, goats, and pigs. With the goal of proposing integrated vector control measures in a One health context, we focused on already used and proven veterinary practices by using at least the species-recommended therapeutic dose of the approved veterinary product, as applied to cure animals from common parasitic diseases.

## Materials and methods

### Ethics approval and consent to participate

This study was conducted in compliance with institutional and national regulations. The study protocol was reviewed and approved by the Institutional Ethics Committee of the "Institut de Recherche en Sciences de la Santé" (IEC-IRSS) and registered as N˚A06/2016/ CEIRES.

### *Anopheles coluzzii* VK5 strain

A colony of one of the major vectors of *Plasmodium*, *An. Coluzzii* [33], was used in this study. The colony was established in year 2008 from 200 wild blood-fed females captured inside houses using a mouth aspirator at the Kou Valley (11˚ 23′14 ″ N, 4˚ 24′42 ″ W) near Bobo-Dioulasso, South-Western Burkina Faso, was used in this study. It is the same than one used for the study published by Pooda et al. 2015 [34], with a proportion of 30–40% mosquitoes carrying the *kdr*-resistant allele conferring resistance to pyretoids. It was repeatedly replenished with F1 from wild-caught mosquito females collected in the same area. The species composition of the colony, its resistance to insecticides status and potential contamination by other species or strains was routinely checked using PCR assay as previously described [34]. The strain was maintained in the "*Institut de Recherche en Sciences de la Santé*" (IRSS) insectary in Bobo-Dioulasso. The insectary' conditions were: a temperature of 27 ± 2˚C, relative humidity of 75 ± 5% and a 12h/12h light/dark photoperiod.

## Experimental animals

This work was carried out using sheep, pigs, and goats purchased from surrounding farms near Bobo-Dioulasso, and transported to the Centre International de Recherche-Développement sur l'Elevage en zones subhumides (CIRDES) experimental barns. The sheep and goats belonged to the Djallonké breed, which is widespread in the study area, while the pigs were of local breed. All animals were dewormed using albendazole (one bolus of 2500 mg of albendazole per animal) one month before the start of the experiment. Additionally, the sheep were treated with a curative trypanocidal (Veriben®) at the prescribed dose of 3.5 mg/kg to clear trypanosome parasites responsible for African Animal Trypanosomiasis, a common disease affecting this species in South-Western Burkina Faso.

The average animal weight at the time of the experiment was 21 ± 3kg for sheep (n = 6), 14 ± 1.65kg for goats (n = 6) and 35 ± 5kg for pigs (n = 8). Water and mineral salts were provided to the animals *ad libitum* throughout the study period. The sheep were fed with rice straw and cottonseed cake at 500 g per day. Three to four kg per day of a concentrate food compound manufactured by the "*Centre de Promotion de l'Aviculture Villageoise*" (CPAVI) and composed mainly of corn, cottonseed cake, and essential amino acids were given to the pigs. Herbaceous legumes (*given at libitum*) and cottonseed cake (500 g per day) were offered to goats.

The injectable veterinary ivermectin formulation (IVOMEC-D®, Boehringer Ingelheim, Lyon, France) was used. This drug contains 1% ivermectin and 10% Clorsulon as active ingredients. IVOMEC-D® targets mainly gastrointestinal nematodes, strongyles, lungworms, and external parasites in cattle and sheep. Clorsulon targets liver flukes. We tested the mosquitocidal effect of blood meals taken by *Anopheles coluzzii* females on hosts treated using the recommended therapeutic dose of the product, which varies according to the animal species: 0.2, 0.3, and 0.4 mg/kg of body weight for sheep, pigs, and goats, respectively. With the pigs, our preliminary data (S1 File) showed only a limited effect of the recommended dose, so two other doses (0.6 mg/kg and 0.9 mg/kg) were also tested. The drug was administered through a single subcutaneous injection into the collier of each animal.

For each species, animals were randomly allocated to the control or experimental groups. The number of animals per group and the number of groups per species are presented in Table 1.

The study was realized sequentially: goats were treated in April 2017, sheep in September of the same year, and pigs in August 2018. Batches of mosquitoes were exposed accordingly.

**Table 1. Distribution of animal species between control and ivermectin-treatment groups.**

| Animal species | Ivermectin treatment (dose, mg/kg) | Number of animals per treatment | Date of injection |
|---|---|---|---|
| Sheep | Ctrl (0) | 3 | 19/09/2017 |
|  | TD (0.2) | 3 |  |
| Goat | Ctrl (0) | 3 | 01/04/2017 |
|  | TD (0.4) | 3 |  |
| Pig | Ctrl (0) | 2 | 28/08/2018 |
|  | TD (0.3) | 2 |  |
|  | 2TD (0.6) | 2 |  |
|  | 3TD (0.9) | 2 |  |

Ctrl: Control; TD: Therapeutic dose.

## Mosquito bioassays

Mosquito blood-feeding bioassays were performed as previously described [34,35]. Briefly, three to five-day-old female *An. coluzzii* mosquitoes were starved for 24 hours before the exposure experiment to increase their appetence for hosts. A circular area was shaved on the flank of each animal to facilitate mosquito feeding. Starved *An. coluzzii* were transferred into plastic cups covered with a net (large radius = 85 mm, small radius = 43 mm, height = 80 mm) and held using a rubber band on the flank of the animal. During the direct skin feeding assays that took place in the morning (between 08:00 a.m. and 11:00 a.m.), animals were restrained using ropes to avoid rough movements and scratching. Mosquitoes were allowed to feed on the shaved animal spots for 30 minutes.

All mosquitoes fed on the same animal were transferred in a large cage from which mosquitoes were individually aspirated using a mouth aspirator and sequentially put in the cups (cup 1 to 4 and then back again to cup 1) until cups were completed to 10 mosquitoes each. All cups were put in trays, and on a shelf, in the insectary. Each day, the cups were taken from the trays, observed for mosquito mortality and put back. To avoid confounding positional effect on mosquito mortality and fecundity phenotypes, trays were rotated from shelf to shelf, and cups inside the trays as well. All the cups were maintained in the same insectary.

## Experimental design for survival, gravidity rate and egg production evaluations

In order to evaluate the effect of ivermectin on survival and fecundity of *An. coluzzii*, the same batches of around 118 (± 65) *Anopheles* were allowed to feed on each treated and control animal before treatment (0 day) and at different time points (2, 7, 14, 21, and 28 days) after injection (DAI). Only fully fed females were considered for survival and fecundity follow-ups. For survival, 40 fed females per each host individual and each time point were randomly distributed in 4 paper cups (ten mosquitoes per cup) and provided every day with cotton balls soaked in a 2.5% glucose solution. Mortality was recorded every day between 8 and 10 a.m. in the morning from the day of blood feeding until all mosquitoes died. Mosquitoes were considered dead if they were lying on the bottom of the cup and unable to move.

Since the first blood meal in *Anopheles gambiae s.l.* mosquitoes is often used to compensate for nutritional deficiencies carried over from larval stages instead of developing ovarian follicles [36,37], the number of eggs produced after two consecutive blood meals is usually considered more representative of actual mosquito fecundity. Thus, for fecundity measurements, remaining female mosquitoes from the first blood feeding were given a second blood meal on the same host 3 days after the first one. Only females that had actually taken two blood meals were further considered. Dissection of the ovaries was performed 4 days later, when the entire digestion of the second blood meal had occurred. Ovaries were extracted from the mosquito abdomen and dissected in a drop of Phosphate-Buffered Saline (PBS) to release the eggs. Eggs were then counted under a binocular (40×, Leica S6D). The proportion of females carrying developed eggs (gravidity rate) and the number of mature eggs (i.e., those that reached Christopher stage V of ovarian development) [38] were considered as proxies of mosquito fecundity, are proxies representing important parameters of the mosquitoes reproductive potential.

## Dosage of ivermectin in animal blood samples

Blood samples from experimental animals were collected in Heparin tubes through the jugular vein using a syringe and a 21-gauge needle. Blood samples were centrifuged at 1500 rpm for 15

min, and the plasma was aliquoted in 1.5 ml Eppendorf tubes and stored at -20˚C until analysis.

Ivermectin extraction and quantification by high-performance liquid chromatography (HPLC) with fluorescence detection after derivatization were performed as previously described [39,40]. First, 1 ml thawed plasma was transferred into a 15 ml polypropylene vial. This was fortified with 25 μl of a 2000 ng/ml doramectin internal standard solution and 3 ml of cold (-32˚C) HPLC-grade acetonitrile (ACN). After ultrasound-assisted extraction and subsequent centrifugation, an aliquot of the supernatant was evaporated under $N_2$ and reconstituted in 600 μl ACN. This was filtered (0.45 μm, polytetrafluoroethylene), and for HPLC-fluorescence detection, ivermectin was derivatized with N-methylimidazole/ACN (1:1, v/v), triethylamine, trifluoroacetic anhydride/ACN (1:1, v/v), and trifluoroacetic acid. Samples were quantified on an Agilent 1200 HPLC system with a C18 column and a gradient elution (purified Milli-Q® water and ACN). Ivermectin concentrations were corrected with doramectin as a surrogate, representing the recovery rate during analysis. Mean IVM recovery (n = 122 samples) ± relative standard deviation in plasma was 86.8 ± 17.6%.

## Statistical analysis

All statistical analyses were performed using the R software, version 1.2.5033 [41].

Blood feeding rate were estimated using general linear models (GLM) using the 'glm' and 'glmmTMB' functions [42] with binomial distribution of the error. Exposed mosquito were used as dependent variable, and treatments, DAI, and interactions were used as explanatory variables.

The survival of *Anopheles* was analyzed using Cox proportional hazard model with treatments, the DAIs (coded as a categorical variable) and interaction as explanatory variables. The model was fitted using the 'coxph' function of the 'survival' package [43]. Theses analyzes were done for each animal blood source.

To assess the individual animal effect on mosquitoes mortality, a Cox model was fitted on data at DAI = 0 (before injection) with individual animals as the explanatory variable.

Gravidity and egg production were estimate using general linear models (GLM) were fitted using the 'glm' and 'glmmTMB' functions [42], respectively, with the binominal and negative binomial distribution of the error. The gravidity status and the number of eggs recorded for each individual mosquito were used as dependent variables, and treatments, DAI, and interactions were used as explanatory variables.

Hazard ratios, odd ratios, rate ratios and their 95% confidence intervals were produced using the 'emmeans' function [44].

## Results

All data supporting these results are available in the supporting information S3–S5 Files.

### Blood feeding rate

There was no effect of the treatment of ivermectin on the rate of mosquitoes blood-fed on sheep ($\chi^2_1 = 0.0867$, $P = 0.77$), goats ($\chi^2_1 = 0.1071$, $P = 0.74$) and on pigs ($\chi^2_3 = 2.5833$, $P = 0.46$), with no significant difference between mosquitoes fed on corresponding treated and control animals (S1 Table). All samples taken together, the rate of blood-fed mosquitoes during the first blood meal was, respectively, 71.52 (±4.88) %, 71.94 (±3.15) % and 57.46 (±2.55) % on sheep, goat and pig (**Fig 1**), and was, for the second blood meal they were respectively 59.57 (±4.55) %, 58.26 (±5.53) %, and 69.46 (±2.74) %, on sheep, goat and pig. The sample size for

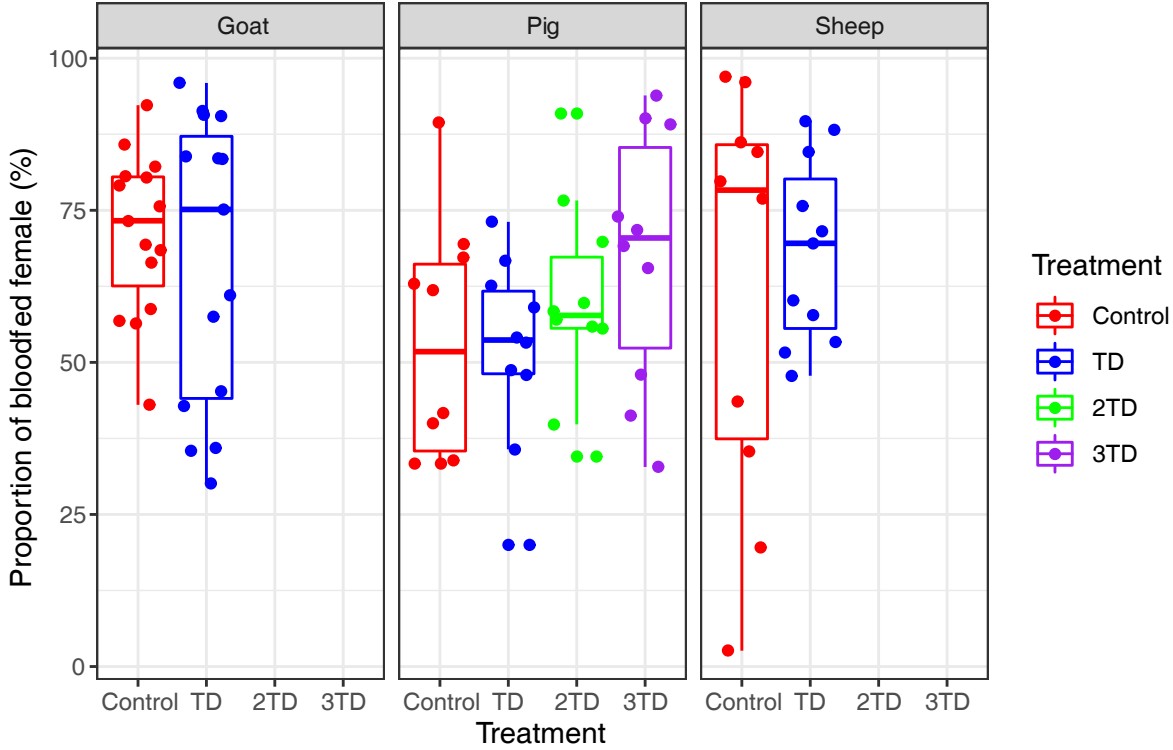

**Fig 1. Rate of blood-fed Anopheles coluzzii according to host specie and ivermectin treatment.** TD = therapeutic dose for the given host species; 2TD = double therapeutic dose; Triple therapeutic dose.

each mosquito group is given in the **S1 Table.** The proportion of bloodfed mosquitoes on pigs is overall lower than on the other animals.

## Survival

The effects of ivermectin on *An. coluzzii* survival were analyzed using a total of 5 590 blood-fed females, with 986, 1 397, and 3 207 female *Anopheles* fed on sheep, goats, and pigs, respectively. Before the treatment, and for each specie, *An. coluzzii* survival was not significantly different between the controls and experimental groups (group effect: sheep, $\chi^2_1 = 7.60$, $P = 0.18$; goats: $\chi^2_1 = 0.003$, $P = 0.96$; pigs: $\chi^2_3 = 2.99$; $P = 0.39$) (**Fig 2**).

Overall, the Cox proportional hazard model showed that ivermectin injected at the therapeutic dose negatively impacted survival of mosquitoes that fed on treated sheep (Treatment effect: $\chi^2_1 = 24.68$, $P < 0.001$), goats (Treatment effect: $\chi^2_1 = 18.34$, $P<0.001$), and pigs (Treatment effect: $\chi^2_1 = 49.80$, $P < 0.001$). The doubled and tripled therapeutic doses used to treat pigs induced as well a significant decrease in mosquito mortality rates (Treatment effect: $\chi^2_1 = 24.85$, $P < 0.001$ for double dose, $\chi^2_1 = 136.46$, $P < 0.001$ for triple dose). Mortality of *An. coluzzii* also significantly varies according to the delay post-ivermectin injection at which blood feeding occurred on treated sheep (DAI effect: $\chi^2_5 = 175.16$, $P<0.001$), goats (DAI effect: $\chi^2_5 = 321.88$, $P < 0.001$) and pigs (DAI effect: $\chi^2_5 = 462.36$, $P<0.001$). Furthermore, the interaction between effects of the treatment and the DAI was significant (treated sheep (Treatment x DAI effect: $\chi^2_5 = 93.67$, $P<0.001$), goats (Treatment x DAI effect: $\chi^2_5 = 35.28$,

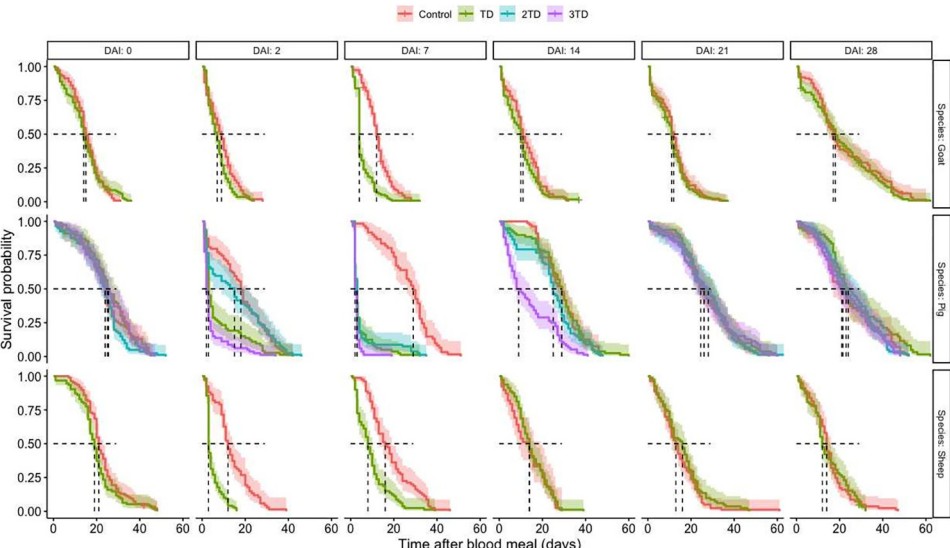

**Fig 2. Kaplan Meir Survival curves of *An. coluzzii* after blood-feeding, displayed by treatment, host species and days after injection.** DAI: Days after injection; TD = therapeutic dose for the given host species.

$P < 0.001$), and pigs (Treatment x DAI effect: $\chi^2_5 = 253.99$, $P < 0.001$). Specifically, the administration of IVM led to a marked reduction in mosquito survival at DAI 2 and 7 when compared to the control group, but there was no discernible difference from DAI 14 to 28.

Accordingly, calculated hazard ratios (HR) showed a significantly higher probability of dying for mosquitoes fed on hosts treated using the therapeutic dose 2 days after injections in sheep (HR = 5.56, IC [4.08–8.90], $P < 0.001$), goats (HR = 1.44, IC [1.11–1.88]; $P = 0.006$) and pigs (HR = 2.42, IC = [1.59–3.68]; $P < 0.001$) (Fig 2). At this timepoint, the median survival time was reduced by 75, 22.2 and 83.0% in mosquitoes fed on treated sheep, goats and pigs, respectively, compared to those fed on control groups.

The effect of the treatment remained significant at 7 DAI in sheep (HR = 2.72, IC [1.99–3.72], $P < 0.001$), goats (HR = 2.95, IC [2.27–3.84], $P < 0.001$) and pigs (HR = 9.35, IC = [5.98–14.64]; $P < 0.001$) (Fig 2). The median survival time at this delay post treatment was reduced by 48.8, 66.7, and 89.7% when female *An. coluzzii* fed on treated sheep, goats and pigs, respectively, compared to mosquitoes fed on the control groups. From day 14 post-treatment and onwards, there was no significant difference between groups whatever the host species considered (Fig 2).

Considering the doubled therapeutic dose (0.6 mg/kg) that was attributed to pigs, *Anopheles* survival was impacted at 7 DAI only (HR = 6.65, IC [4.28–10. 34]; $P < 0.001$; Fig 2), with an 89.7% decrease in female survival time when fed on treated pigs compared to the controls. Treatment with the triple therapeutic dose (0.9 mg/kg) provoked significant higher mortality in mosquitoes lots fed at 2 DAI (HR = 4.98, IC [3.30–7.53], $P < 0.001$) and 7 DAI (HR = 15.75, [9.94–24.94], $P < 0.001$), and extended the mosquitocidal effect until 14 DAI (HR = 2. 71, IC [1.79–4.10]; $P < 0.001$). Compared to mosquitoes fed on control pigs, the decrease in survival was 88.89%, 93.10% and 68.97% for *An. coluzzii* mosquitoes fed on treated ones (data for DAI of 2, 7 and 14, respectively).

## Gravidity

Overall, 2,373 gravid females *An. coluzzii* were dissected after two successful blood meals, with 652 and 938 females fed on sheep and goats, respectively. Sample sizes obtained for pigs did

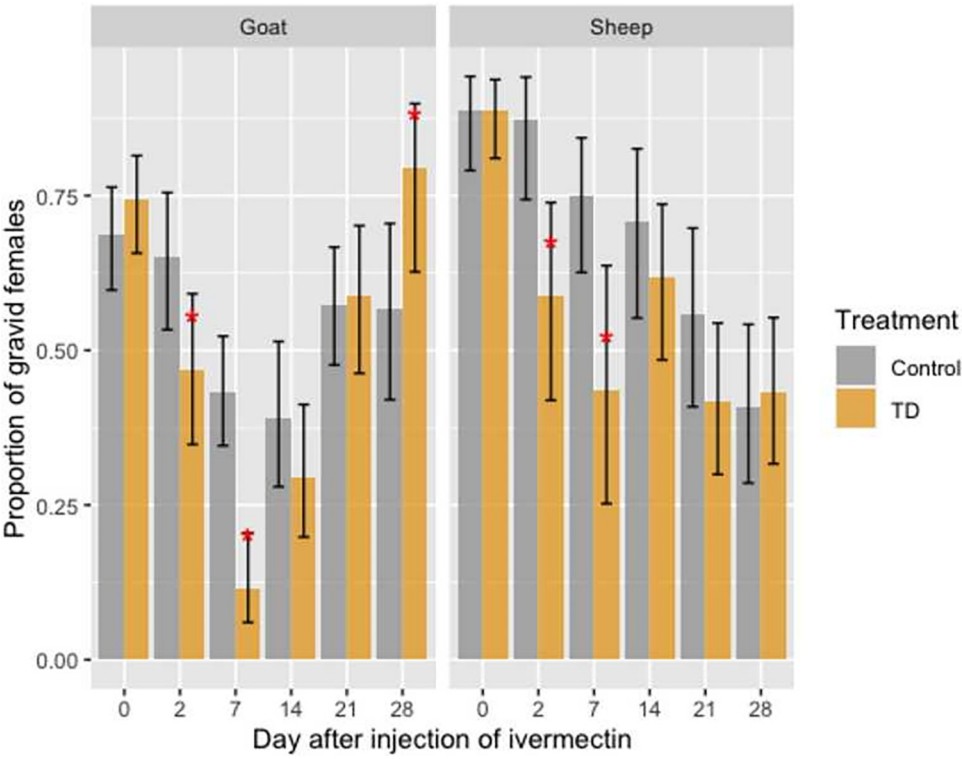

**Fig 3.** Gravidity rates of female *An. coluzzii* after two successful blood meals on control (gray) and ivermectin-(orange) treated sheep and goats at day(s) 0, 2, 7, 14, 21, and 28 post-treatments. Error bars correspond to the Standard Error.

not allow further analysis. (**S1 Table and S2 File**). Gravidity rates (proportion of females carrying developed eggs) according to the blood meal origin and DAI are shown in **Fig 3**.

Before the treatment, gravidity rates of females *An. coluzzii* were similar between the control and treated groups of sheep and goats (all $P > 0.05$).

Treatment of sheep and goats with their respective therapeutic dose of ivermectin had a significant impact on *An. coluzzii* gravidity rate. Indeed, for *An. coluzzii* that fed on treated sheep, gravidity rates were reduced, compared to those fed on controls, by 32.57 and 42.03% at 2 DAI (OR = 4.78, IC [1.60–2.80], $P = 0.005$) and 7 DAI (OR = 3.90, IC [1.42–10.71], $P = 0.008$) respectively. Similarly, treatment of goats with ivermectin led to a 28.28 and 73.64% reduction of *An. coluzzii* gravidity rate in treated group compared to control at 2 DAI (OR = 2.13, IC [1.06–4.31], $P = 0.03$) and 7 DAI (OR = 5.92, IC [2.70–12.96], $P < 0.001$) respectively. No significant decreasing effect of treatment was observed at 14, 21 or 28 DAI (**Fig 3**). However, an unexpected 39.76% significant increase in gravidity rate was observed in *An. coluzzii* that fed on treated-goats at 28 DAI compared to those fed on control animals (79.41% ± 6.93% vs 56.82% ± 7.46% in the control group, **Fig 3,** OR = 0.34, IC [0.123–0.949], P = 0.04)).

## Development of mature eggs

Before the injection of ivermectin, the mean number of developed mature eggs by female *An. coluzzii* that fed on control and treated sheep (140.17 vs. 145.13 eggs/female) and goats (161.02 vs. 147.47 eggs/female) was not significantly different (**Fig 4**).

In sheep, a significant decrease of the mean number of eggs developed by female was observed at 28 DAI only, with a mean number of eggs decreasing from 137.4 (±16.8) eggs/

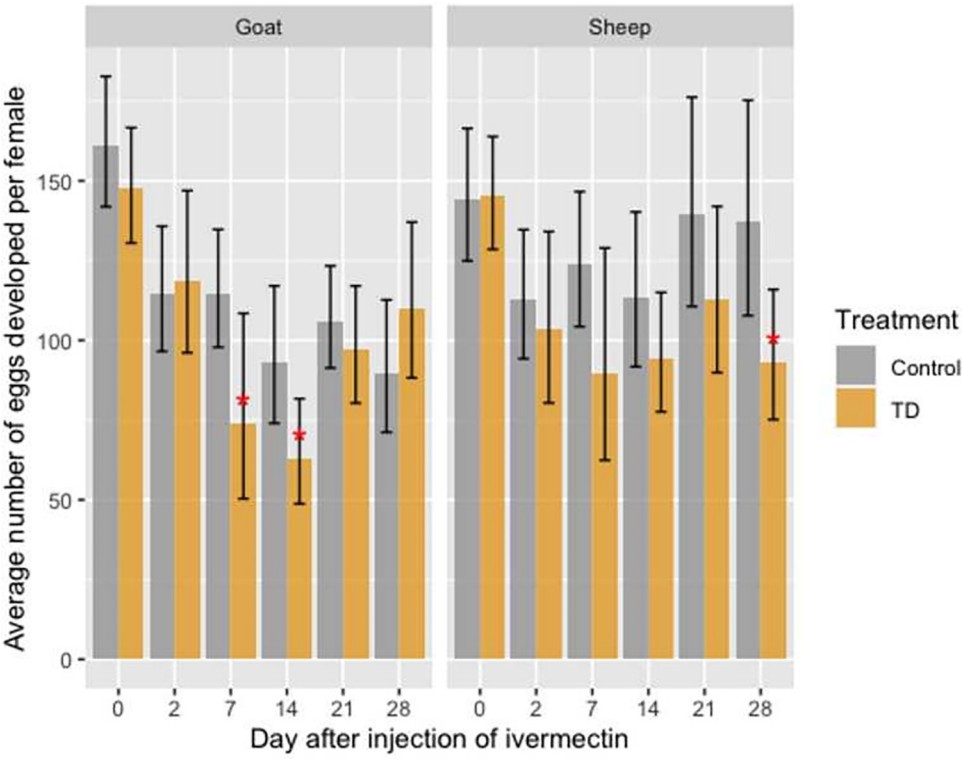

**Fig 4. Average number of developed eggs by gravid female *An. coluzzii* according to blood meal origin (sheep or goat), DAI and treatment.** Error bars correspond to the Standard Error.

female in the control group to 93.4 (±10.15) eggs/female in the treated group (**Fig 4**). This represents a 32.06% decrease in egg production (OR = 1.45, IC [1.07–2.03], $P$ = 0.02). A similar trend was observed at 7, 14, and 21 DAI, but was not significant.

In goats, injection of ivermectin at a dose of 0.4 mg/kg led to a significant decrease in the mean number of eggs developed by a female *An. coluzzii* fed on treated animals compared to controls at 7 DAI (OR = 1.55, IC [1.02–2.37]; $P$ = 0.04) and 14 DAI (OR = 1.48, IC [1.04–2.09]; $P$ = 0.03), with a 35.7% and 32.2% decrease in egg production, respectively (**Fig 4**).

### Ivermectin plasma concentration in treated animals

**Table 2** shows the mean plasma concentrations of ivermectin in sheep, goats, and pigs according to treatment and DAI. The single timepoint at which ivermectin concentration was assessed in sheep precludes any consideration of the concentration dynamics for this species. Analyses revealed that plasma concentrations in goats and pigs were relatively constant and reached a peak between day 2 and day 7 post-treatment. Following that, ivermectin concentrations decreased toward low or undetectable levels, depending on the treated host and treatment dose. Interestingly, ivermectin plasma concentrations were 2 to 5-fold higher in pigs compared to goats at all timepoints, regardless of the dose (**Table 2**). For pigs in particular, a great inter-individual host variability in ivermectin plasma concentration can be noticed.

### Discussion

The presence of alternative blood sources (*i.e.* cattle, sheep, goats, pigs, dogs . . .) to human allows zoophagic and opportunistic *Anopheles* mosquitoes to maintain and continue to

**Table 2. Median plasma concentrations of ivermectin (ng/mL) in the treated- sheep, goats and pigs according to the DAI.**

| Species | Dose [mg/kg] | Median (Min–Max) plasma concentration of ivermectin (ng/ml) according to the DAI | | | |
|---|---|---|---|---|---|
| | | 2 | 7 | 14 | 21 |
| **Goat (n = 3)** | 0.40 | 6.4 (0–7.8) | 9.00 (6.7–12.4) | 0.0 (0–3.7) | - |
| **Pig (n = 2 per Dose)** | 0.30 | 22.50 (11.7–33.3) | 20.65 (19.5–21.9) | 6.5 (6.2–6.8) | 0 |
| | 0.60 | 20.35 (7.6–33.1) | 27.55 (23.3–31.8) | 12.25 (6.2–18.3) | 2 (0–4) |
| | 0.90 | 40.45 (32.9–48) | 33.85 (32.6–35.1) | 10. 7 (9.5–11.9) | 4.95 (3.8–6.1) |
| **Sheep (n = 3)** | 0.20 | - | 6.00 (5.7–6.3) | - | - |

transmit malaria despite high population coverage using LLINs and IRS [27,45,46]. However, this feeding behavior may provide an opportunity for zooprophylaxis and to controlling these malaria vectors using the insecticide-treated livestock (ITL) strategy [13]. Herein, we investigated the impact of ivermectin-treated sheep, goats and pigs on the survival and the production of mature eggs of *An. coluzzii*, major vector of *Plasmodium* throughout western Africa. The results demonstrated that ivermectin treatment of each animal species at the therapeutic dose reduced the survival and eggs production of *An. coluzzii*. The survival of *An. coluzzii* was decreased by 89.7, 66.7, and 48.4% when fed on treated pigs, goats, and sheep, respectively at 7 DAI, comparatively to those fed on untreated animals. In addition, female *An. coluzzii* that did survive after feeding on treated animals exhibited a reduced number of eggs developed (32% reduction for mosquitoes that fed on sheep at 28 DAI and 35.65% and 32.21% for those fed on goats respectively at 7DAI and 14 DAI). At therapeutic doses, toxicity to survival seemed higher in pigs compared to goats and sheep, although significance of this comparison is loose because studies in pigs were performed at separate time periods, using different mosquito batches. However, ivermectin plasma concentration measurements clearly revealed that ivermectin availability in the host bloodstream differed between species. This is especially evident when comparing pigs and goats, with the former displaying the highest ivermectin plasma level at all considered time-points although they received the lowest therapeutic dose. Our results are consistent with those reviewed by Alvinerie and al. [47] and they may be explained by species-specific parameters that impact the pharmacokinetic properties of ivermectin in a given organisms, as the absorption process, the volume of distribution of the molecule, the different body compartments where the molecule is actually distributed, the lean *vs* fat weight as ivermectin is highly hydrophobic, the metabolism of the considered species and the metabolization of ivermectin into secondary metabolites [47–49]. These parameters actually determine quantitatively and qualitatively the molecule's distribution in the different body compartments including the skin's blood capillaries where the mosquitoes bite. Our data show that mosquitocidal efficacy for a given ivermectin plasma concentration differs from a species to another and that inter-species extrapolations of efficacy or effectiveness based on plasma ivermectin concentrations only are not meaningful. Lethal plasma concentrations should therefore be estimated at the least for each species.

Differences in pharmacokinetics properties among species is also illustrated by differential duration of the lethal effect of the therapeutic dose, which is 7 days in our study for sheep, goats and pigs, but is longer lasting for cattle (3 weeks post-treatment, shown suing the same experimental approach [34]).

Mosquitocidal efficacy and plasma concentrations data in pigs receiving the therapeutic dose are in line with previous studies [20]. Increasing treatment doses did not lead to a proportional rise in median plasma ivermectin concentrations, nor in efficacy. For this species, significant data variability was observed among animals within the same treatment groups, and the small number of pigs per group clearly limits our study. However, this same design has also been used in other studies [20,35].

The proportion of blood-fed mosquitoes was lower in pigs compared to sheep and goats (**Fig 2**). This may be due to thicker skin and increased agitation and skin tremors observed in pig during feeding assays, which can be seen as defensive behavior against blood-feeing ecto-parasites like *Anopheles* [50]. This behavior could lead to less effective blood meals, although our observations of engorged females did not show any noticeable differences. However, in the field, the proportion of *Anopheles coluzzii* feeding on pigs might be higher since defensive behaviors are naturally reduced at night, when main *Plasmodium* vectors are active.

Sub-lethal ivermectin plasma concentrations also impaired gravidity rate and fecundity, expressed as the number of eggs developed in mosquitoes ovaries, when female *Anopheles* blood fed on ivermectin-treated sheep and goats. Such reduction in fecundity potential is in line with numerous previous studies [21,34,51]. These sub-lethal effects on vectors reproductive outputs would, in the fields, translate into greater and/or prolonged efficacy at decreasing *Anopheles* densities, and woul sustain decreasing *Plasmodium* transmission. Current models used to predict ivermectin mass administration impacts on *Plasmodium* transmission and malaria epidemiology focus on mosquito survival data only and, therefore, probably underestimate the ability of this drug to be a powerful complementary vector control tools. Taking into account reproductive outputs in future models would allow a more complete comprehensive projection of potential impacts of ivermectin-based zooprohylaxy approach.

Our present and past results [34] demonstrated that ivermectin treatments of the main livestock species in rural endemic Africa significantly reduce the fitness of *Anopheles* vector under laboratory conditions. Scaling-up such approach in the field will require a better understanding of the molecule pharmacokinetics and pharmacodynamics for all the *Anopheles* host species, as well as a better characterization of the blood feeding preference of these vectors in the fields. Innate feeding preferences does not necessarily reflect the actual realized bloodmeal [7,27], which depends on the relative proportions of the different host and their accessibility to mosquitoes. The proportion of opportunistic mosquitoes from major vectors species is likely underestimated in the fields, so is the role domestic animals play in maintaining their populations and the residual *Plasmodium* transmission despite optimal control tools deployment. Moreover, bloodmeals origin assessments suffer from sampling challenges of outdoor feeding mosquitoes, a feeding trait often associated with zoophagy [13]. In Burkina Faso, rural communities live in close proximity to cattle, as well as sheep, goats, and pigs [52]. Cattle are the preferred alternative hosts for *An. gambiae s.l.* populations [7,27], but the number of sheep and goats can far exceeds that of cattle, potentially leading to more mosquitoes targeting these animals as alternative hosts. Pigs are primarily found in the southwestern region, where their densities are the lowest compared to the other species. Therefore, in areas where ivermectin-based interventions are planned, it is essential to thoroughly assess blood indexes in malaria mosquito populations, conduct a census of the different host species, and perform species-specific pharmacokinetics and pharmacodynamics studies to predict effectiveness accurately. This also involves determining the duration of significant mortality effects for each mosquito species.

The effectiveness of this approach also depends on timely interventions, particularly at the onset of the rainy season to prevent mosquito populations from increasing exponentially [53], and on using appropriate treatment schemes and doses. This includes utilizing long-lasting technologies to maintain ivermectin blood concentrations at effective levels throughout the Plasmodium transmission season [35,54,55], while easing the logistics of multiple dosing schemes.

The treatment coverage and overall implementation of the approach will inevitably be constrained by the ivermectin usage guidelines established by the Joint FAO/WHO Expert Committee on Food Additives [56], particularly concerning milk and slaughter withdrawal periods.

The benefits to animal health and the long-term wealth of herders must be balanced against potential short-term resource shortages. Therefore, the number of animals to be treated should be determined in consultation with herders, using integrative models to ensure that effectiveness is achieved. Interestingly, not treating entire herds will create refugia for susceptible endo and ectoparasites including Anopheles vectors, providing mitigation strategy against ivermectin resistance [57,58]. This same constraint also provides refugia for non-targeted fauna including coprophagic organisms. However, environmental risk assessments should be conducted, and mitigation measures implemented, to ensure the sustainability of this approach and to protect already fragile ecosystems and agro-ecosystems, where manure plays a crucial role in soil fertilization [55,59].

## Conclusions

The present study shows that ivermectin-treated sheep, goats, and pigs, using the prescribed dose for each species, reduce the survival and fecundity potential of the opportunistic malaria vector *Anopheles coluzzii* under laboratory conditions. However, the limited period of efficacy questions the practical feasibility of using this veterinary formulation in the fields, because the logistical and cost issues for providing sufficient coverage to span the whole transmission season. Overcoming these limitations will require the use, for instance, of slow-release technologies that allow sustained efficacy for more than a month, as recommended by the WHO for endectocides approaches in malaria vector control. Such an integrated intervention, in the One-Health frame, would address the so far neglected aspect of *Plasmodium* transmission that involves domestic animals. Opportunistic, major malaria vectors feed on these readily accessible alternative hosts, which allow residual transmission to occur despite high control tool coverage. Endemic countries carrying the highest malaria burden are in their vast majority also countries where human and domestic animal populations lives are intricated. Such an approach, if environmental concerns are also addressed, will likely allow such malaria endemic countries to get closer from their development goals in terms of both health and wealth.

## Supporting information

**S1 Table. Number of exposed, blood-fed and dissected female *Anopheles* according to the hosts species, the treatment and the time elapsed since the injection (DAI).**
(PDF)

**S1 File. Preliminary study: Efficacy of blood meal taken on treated pigs with therapeutic dose of ivermectin on the survival of *Anopheles coluzzii*.**
(PDF)

**S2 File. Gravidity rate and fecundity in females *Anopheles* coluzzii fed on pig.**
(PDF)

**S3 File. Data set on the bloodmeal rate of *Anopheles coluzzii*.**
(TXT)

**S4 File. Data set on the survival of *Anopheles coluzzii*.**
(TXT)

**S5 File. Data set on the gravidity status and the number of eggs developed in *Anopheles coluzzii*.**
(TXT)

## Acknowledgments

This paper is in Memory of Jean-Baptiste Rayaissé and Issa Sidibé who were one of the main actors of this work and who passed away! Rest in peace!

We thank sincerely the Merial laboratory which graciously gave us the ivermectin-based medicine, as well as all the technicians and animal keepers who helped us in carrying out this work.

## Author Contributions

**Conceptualization:** Sié Hermann Pooda, Roch Kounbobr Dabiré, Amnan Alphonsine Koffi, Rolf-Alexander Düring, Cédric Pennetier, Nicolas Moiroux, Karine Mouline.

**Data curation:** André Barembaye Sagna, Thierry Lefèvre, Nicolas Moiroux, Karine Mouline.

**Formal analysis:** Sié Hermann Pooda, Andre Patrick Heinrich, Angélique Porciani, Karine Mouline.

**Investigation:** Sié Hermann Pooda, Domonbabele François de Salle Hien, Soumaïla Pagabeleguem, Andre Patrick Heinrich, Lamidi Zela, Lassane Percoma, Karine Mouline.

**Methodology:** Sié Hermann Pooda, Karine Mouline.

**Supervision:** Roch Kounbobr Dabiré, Rolf-Alexander Düring, Cédric Pennetier, Nicolas Moiroux, Karine Mouline.

**Validation:** Karine Mouline.

**Writing – original draft:** Sié Hermann Pooda.

**Writing – review & editing:** Sié Hermann Pooda, Domonbabele François de Salle Hien, Soumaïla Pagabeleguem, Andre Patrick Heinrich, Angélique Porciani, André Barembaye Sagna, Lamidi Zela, Lassane Percoma, Thierry Lefèvre, Roch Kounbobr Dabiré, Amnan Alphonsine Koffi, Rolf-Alexander Düring, Cédric Pennetier, Nicolas Moiroux, Karine Mouline.

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
