## [Decision Letter · Decision Letter 0]

3 Jan 2024

PONE-D-23-39281Impact of blood meals taken on ivermectin-treated livestock on survival and fecundity of the malaria vector Anopheles coluzzii under laboratory conditionsPLOS ONE

Dear Dr. Pooda,

Thank you for submitting your manuscript to PLOS ONE. After careful consideration, we feel that it has merit but does not fully meet PLOS ONE’s publication criteria as it currently stands. Therefore, we invite you to submit a revised version of the manuscript that addresses the points raised during the review process. Both reviewers did a detailed analysis with several suggestions that will help to improve your manuscript. Please give special attention to the questions on the colony's insecticide resistance status and the lack of correlation between ivermectin plasma concentrations and mortality.

We look forward to receiving your revised manuscript.

Kind regards,

Pedro L. Oliveira

Academic Editor

PLOS ONE

A clean copy of the edited manuscript (uploaded as the new *manuscript* file)".

Reviewers' comments:

Reviewer's Responses to Questions

**Comments to the Author**

1. Is the manuscript technically sound, and do the data support the conclusions?

Reviewer #1: Yes

Reviewer #2: Yes

2. Has the statistical analysis been performed appropriately and rigorously? 

Reviewer #1: Yes

Reviewer #2: Yes

3. Have the authors made all data underlying the findings in their manuscript fully available?

Reviewer #1: Yes

Reviewer #2: Yes

4. Is the manuscript presented in an intelligible fashion and written in standard English?

Reviewer #1: Yes

Reviewer #2: Yes

5. Review Comments to the Author

Reviewer #1: Pooda and Colls investigated the effects of ivermectin treatments as an ectocide in three livestock species (Sheep, goats and pigs) on the survival and reproduction of the malaria vector Anopheles coluzzii. Different domestic animal species have been used since the pharmacokinetics of Ivermectin may vary between vertebrate hosts. This strategy could be used to complement current control tools such as LLINs and IRS to target exophagic, exophilic and zoophagic vectors. The injectable veterinary ivermectin formulation at the species-specific doses caused a significant decrease in mosquito survival for up to 7 days after injection. The number of gravid females Anopheles that survived after feeding on treated animals was also reduced, as well as the number of mature eggs in the ovaries. However, due to the short-term efficacy of single-dose treatments, repeated treatments and potentially increased dosages would be required to span the transmission season.

The methodology seems to be appropriate, and the results obtained support the conclusions realised by the authors. However, some issues and concerns should be addressed.

1. The authors considered the proportion of females carrying eggs (gravidity rate) and the number of mature eggs in the ovaries as proxies of mosquito fecundity. It is not the optimal way to assess the effect of a drug on reproductive fitness. The presence of a drug may delay ovaries and egg development without affecting the final reproductive output. A better and more direct way to assess reproductive fitness would be to allow the treated females to complete the reproductive cycle, lay the eggs, and count the number of eggs laid by each female and their hatching rate. The final number of F1 per female is the better way to quantify the effect of a drug on reproductive fitness.

2. Which is the insecticide resistance status of the An coluzzii colony used? Please add this information, if known.

3. ¨Lines 235-236: ¨The rate of blood-fed mosquitoes was, respectively, 71.52 (±4.88) %, 71.94 (±3.15) % and 57.46 233 (±2.55) % on sheep, goat and pig at the first blood meal (Figure 1)¨.

Were these values calculated using both control and Ivermectin-treated animals? Please clarify this issue.

4. Figure 1: Change DT for TD in the X-axis information. What’s the meaning of ten in the X-axis legend? This figure is difficult to interpret and contains some mistakes. I guess the pink columns are the insects fed on control animals, blue are mosquitoes fed on therapeutical dose-treated animals (TD), while green and violet colours are 2TD and 3DT in mosquitoes fed on pigs. Please clarify this and accommodate the order of columns in panel C according to panels A and B. Add ¨3TD¨ at figure legend.

5. Line 255-256: ¨The doubled and tripled therapeutic doses used to treat pigs induced as well a significant decrease in mosquito mortality rates¨

Should be ¨ a significant increase¨

6. Figure 2: There seems not to be differences in survival in IVM-treated and control goats on day 2 DPI. Please check it. Besides, the mortality in pigs is higher for TD than for 2TD on day 2 DPI. Please also check this issue. Add the time units in the X-axis (days?)

7. Line 282-283: From day 14 post-treatment and onwards, there was no significant difference between groups whatever the host species considered (Figure 1).

Should be ¨ Figure 2¨

8. Figures 3 and 4: Please, add asterisks to the statically significant differences between control and IVM-treated animals.

9. There is no correlation in the mean plasma concentrations of ivermectin (ng/mL) in the treated- sheep, goats and pigs (table II) with the mortality observed in Figure 2 and the number of eggs in Figure 4. Can the authors explain this? For example, in pigs, only the 3TD caused increased mortality on day 14 DPI, but the concentration is as high as 2TD, which did not increase mortality on that day. Besides, 2TD on pigs doubles the concentration in goats at 2 DPI and in sheep at 7 DPI, and both increased mosquitoes mortality. A deeper discussion about the lack of correlation in plasma IVM concentrations and the phenotypes observed associated with reduced survival and reproductive fitness is needed. The lack of correlation makes the interpretation of the results very difficult.

Reviewer #2: PONE-D- 23-39281

Impact of blood meals taken on ivermectin-treated livestock on survival and fecundity of the malaria vector Anopheles coluzzii under laboratory conditions

This review is by Carlos Chaccour from ISGlobal, Barcelona Institute for Global Health. I have a personal open peer-review policy as the current single-blinded system is riddled with vices.

This manuscript reports the results of an experiment conducted in Burkina Faso in which pigs, sheep and goats were treated with different doses of ivermectin and then An. coluzii mosquitoes were fed on them at different times after treatment. The authors also collected some PK data. The results are discussed on the context of a potential One Health approach to malaria control.

Albeit some minor mistakes, the manuscript is well written. The methods are appropriate for the objectives and the conclusion is supported by the result. I provide here comments for the author`s consideration.

Introduction

Consider mentioning early exit as another mosquito behavior contributing to residual transmission. This can be induced by the repellent properties of indoor insecticides or occur even in their absence.

There is no mention of two key concepts: zooprophylaxis vs zoopotentiation.

Methods

Please mention the colony`s insecticide resistant status. This is important given the potential cross-metabolic resistance with pyrethroids as they share the same CYP as ivermectin.

Please mention the calculated adipose vs lean weight of the animals.

How was the random allocation of mosquito cups done?

Were cups rotated in the insectary?

Figure 2 is pixelated, making reading it difficult. Additionally, the X axis seems compressed, giving the illusion that the curves are smooth rather than the usual K-M step by step drop. I recommend providing a higher quality image and perhaps even separating it in three different figures to ensure sufficient size. Consider also adding guiding marks to the reader such as the median survival in the control group.

Results

The increased gravity found at 28 DAI in goats does not seem to be statistically significant as the CI overlap in figure 3. If that is the case, I recommend, stating it in the text.

The same for the decrease in fecundity reported at day 28 in sheep, or 7 DAI in goats, although in these cases p-values are provided in the text, it is worth it to double check the figure given that CI-overlap.

Table II. The metric should be median and range given the samples come from only two animals.

Can the authors use the PK data to estimate the 7-day LC50?

What hypotheses do the authors have about the disparity between the ivermectin concentrations and the mosquito mortality seen in pigs?

There is no mention of toxicity in the livestock. Did the authors monitor for toxicity in pigs given three-fold doses?

Discussion

Please comment in the expected relative densities of each livestock species in the field. Are goats more common than cattle? What order of livestock treatment would you recommend? Cattle > Pigs > Goat > Sheep? Or other?

There is no mention about the milk or slaughter withdrawal periods and how this may affect deployment of the proposed strategy.

Please consider mentioning the potential impact of intense treatment schemes or long-lasting formulations on intestinal parasites resistance. What role could refugia play?

There is also no mention about the potential long-term theoretical risk of selectin a more anthropophilic mosquito population.

Minor

Lines 482. Passed away. Not “are passed away”. Also Rest in peace not “rested in peace”

6. PLOS authors have the option to publish the peer review history of their article (what does this mean?). If published, this will include your full peer review and any attached files.

Reviewer #1: **Yes: **Marcos Sterkel

Reviewer #2: **Yes: **Carlos Chaccour

---

## [Author Response · Author response to Decision Letter 0]

1 Jul 2024

PONE-D- 23-39281

Impact of blood meals taken on ivermectin-treated livestock on survival and fecundity of the malaria vector Anopheles coluzzii under laboratory conditions

Responses to Editor:

Response: These requesting has been taken account.

Response: Information about the permits obtained for the work has been added to the Materials and Methods’ section of the revised manuscript (lines 118–121).

Response: These requesting has been taken account. The co-authors edited the manuscript.

Response: This has been corrected in the revised manuscript (lines 492-495).

Response: The ethics statement has been moved to the Materials and Methods section as suggested (lines 118–121).

Response: The changes have been made in the Supporting Caption on lines 476-481.

 

Response to the Reviewer #1:

Pooda and Colls investigated the effects of ivermectin treatments as an ectocide in three livestock species (Sheep, goats and pigs) on the survival and reproduction of the malaria vector Anopheles coluzzii. Different domestic animal species have been used since the pharmacokinetics of Ivermectin may vary between vertebrate hosts. This strategy could be used to complement current control tools such as LLINs and IRS to target exophagic, exophilic and zoophagic vectors. The injectable veterinary ivermectin formulation at the species-specific doses caused a significant decrease in mosquito survival for up to 7 days after injection. The number of gravid females Anopheles that survived after feeding on treated animals was also reduced, as well as the number of mature eggs in the ovaries. However, due to the short-term efficacy of single-dose treatments, repeated treatments and potentially increased dosages would be required to span the transmission season.

The methodology seems to be appropriate, and the results obtained support the conclusions realised by the authors. However, some issues and concerns should be addressed.

Response: We thank sincerely Dr Marcos Sterkel for the reading and analyzing our paper. We appreciated greatly your observations made about this work.

1.The authors considered the proportion of females carrying eggs (gravidity rate) and the number of mature eggs in the ovaries as proxies of mosquito fecundity. It is not the optimal way to assess the effect of a drug on reproductive fitness. The presence of a drug may delay ovaries and egg development without affecting the final reproductive output. A better and more direct way to assess reproductive fitness would be to allow the treated females to complete the reproductive cycle, lay the eggs, and count the number of eggs laid by each female and their hatching rate. The final number of F1 per female is the better way to quantify the effect of a drug on reproductive fitness.

Response: We agree that using only the number of mosquito females carrying eggs and the number of mature eggs per female gives only part of the answer on the impact of ivermectin on the overall reproductive fitness of Anopheles coluzzii. We are however confident that we are not observing a delay in ovaries and egg development and that the number of eggs we counted by gravid female represents the final output in terms of the maximum number of eggs that a female could carry, since we let the female to develop eggs for 4 days post blood-meal, and since during dissection, all the eggs observed and counted under a binocular were mature (Christopher stage 5). Moreover, no remaining, undigested blood, has been observed. We are however aware about the fact that these counts (% gravids and number of eggs per female) represent only a “potential” and that the actual number of laid eggs and, moreover, the number of larvae that develop from these eggs illustrate better the actual reproductive fitness. Although other authors used as well the number of eggs counted through dissection as a fecundity index (Mekuriaw et al. Malar J (2019) 18:357 https://doi.org/10.1186/s12936-019-2988-3), we took into consideration the reviewer’s remark and modified several sentences accordingly throughout the text, including the manuscript title (fecundity was replaced by eggs production). We directly addressed this issue in the materials and methods section, lines 207-210, by acknowledging the fact that we present a proxy of the fecundity expressing a potential only.

“The proportion of females carrying developed eggs (gravidity rate) and the number of mature eggs (i.e., those that reached Christopher stage V of ovarian development) [39] are proxies representing important parameters of the mosquitoes reproductive potential.”

2.Which is the insecticide resistance status of the An. coluzzii colony used? Please add this information, if known.

Response: The colony we used during our experiments in 2017 and 2018 was the same as in the study from Pooda et al. 2015. Like stated in this article, the colony was repeatedly replenished with wild mosquitoes from the village of Bama, where founder individuals were collected as well and where the mutated kdr allele prevalence is very high. We therefore assume that we also dealt with mosquito batches displaying the same proportion of pyrethroid resistant mosquitoes than previously reported, which was 30-40%. This information is now added in the manuscript lines 123-134.

“A colony of one of the major vectors of Plasmodium, An. coluzzii, was used in this study. The colony was established in year 2008 from 200 wild blood-fed females captured inside houses using a mouth aspirator at the Kou Valley (11 ° 23′14 ″ N, 4 ° 24′42 ″ W) near Bobo-Dioulasso, South-Western Burkina Faso, was used in this study. It is the same than the one used for the study by Pooda et al. (2015), with a proportion of 30-40% mosquitoes carrying the kdr-resistant allele conferring resistance to pyrethroids. It was repeatedly replenished with F1 from wild-caught mosquito females collected in the same area. This Anopheles species is one of the major vectors of Plasmodium parasites in Burkina Faso [33]. The species composition of the colony, its resistance to insecticides status, and potential contamination by other species or strains was routinely checked using PCR as previously described [34] »

3. Lines 235-236: The rate of blood-fed mosquitoes was, respectively, 71.52 (±4.88) %, 71.94 (±3.15) % and 57.46 233 (±2.55) % on sheep, goat and pig at the first blood meal (Figure 1).

Were these values calculated using both control and Ivermectin-treated animals? Please clarify this issue.

Response: The values presented here were calculated with the overall data considering mosquitoes fed on control and treated animals. This is because there were no significant differences between treatments. Details of the number of mosquitoes fully engorged for each group of animals and at all considered time points are given in the Supplementary Table S1. The reviewer’s comment has been taken into account and the reported sentence has been completed as follows, lines 248-255: 

“There was no effect of the treatment of ivermectin on the rate of mosquitoes blood-fed on sheep (χ21=0.0867, P = 0.77), goats (χ21=0.1071, P = 0.74) and on pigs (χ23=2.5833, P = 0.46), with no significant difference between mosquitoes fed on corresponding treated and control animals (S1 Table). All samples taken together, the rate of blood-fed mosquitoes during the first blood meal was, respectively, 71.52 (±4.88) %, 71.94 (±3.15) % and 57.46 (±2.55) % on sheep, goat and pig at the first blood meal (Figure 1), and was, for the second blood meal, 59.57 (±4.55) %, 58.26 (±5.53) %, and 69.46 (±2.74) %. The sample size for each mosquito group is given in the supporting S1 Table”.

4. Figure 1: Change DT for TD in the X-axis information.

Response: This has been done, see the Figure 1

What’s the meaning of ten in the X-axis legend? This figure is difficult to interpret and contains some mistakes. I guess the pink columns are the insects fed on control animals, blue are mosquitoes fed on therapeutical dose-treated animals (TD), while green and violet colours are 2TD and 3DT in mosquitoes fed on pigs. Please clarify this and accommodate the order of columns in panel C according to panels A and B. Add ¨3TD¨ at figure legend.

Response: We thank the reviewer for his remarks to improving the quality of our figure. However, we are quite puzzled because they do not seem to fit if we consider the Figure 1 that we believe we submitted together with our manuscript (see below). We are confused and we very much apologize because the mistake must come from our side with a preliminary instead of final Figure 1 submitted. We hope that the reviewer will find the actual Figure 1 below being suitable for illustrating our results about the proportion of Anopheles females that fed on hosts of different species treated with different ivermectin doses. As for answering to the question “What’s the meaning of ten in the X-axis legend?”, there is no more “10” in the X-axis legend.

Response: The corrected figure 1 is uploaded. 

5. Line 255-256: The doubled and tripled therapeutic doses used to treat pigs induced as well a significant decrease in mosquito mortality rates

Should be ¨ a significant increase

Response: This has been corrected in the manuscript. This reads now at lines 298 - 306.

6. A. Figure 2: There seems not to be differences in survival in IVM-treated and control goats on day 2 DPI. Please check it. 

Response: Thank you for your observations. For this specific experimental point at 2DAI, the statistical analyses showed a significant decrease in Anopheles survival between IVM-treated and control goats (HR = 1.46, IC [1.09 – 1.95]; P= 0.01). The effect is not graphically obvious due to the duration of the follow up per se (until all mosquitoes died) but also, an unexpected low survival rate in the control arm occurred at this DAI. However, this result is statistically supported (lines: 281-286). The median survival at 2 DAI was 9 days for the control mosquitoes (8, 8, and 9 days for the three goats, respectively) and 7 days for mosquitoes that fed on treated goats (7, 6, and 7 days for the three IVM-treated goats, respectively).

6. B. Besides, the mortality in pigs is higher for TD than for 2TD on day 2 DPI. Please also check this issue. 

Response: To better visualize mortality data we obtained when exposing mosquitoes to control and treated pigs 2 days after injection of ivermectin, we plotted below the survival curves for this DAI and for the different treatments. We can notice the variability of the toxic effect generated by blood feeding on the pigs P3 and P6 from the 2TD, and also the TD treatments. Such variability is directly linkable to the variability observed in ivermectin concentration values (see table 2 below).

Figure 2. Survival curves for mosquitoes from the same lot fed on control (P1 and P8) and treated pigs 2 DAI after treatments. Ivermectin treatments were injected at the therapeutic dose (TD, P2 and P7), the double (2TD, P3 and P6) and the triple (3TD, P4 and P5) therapeutic dose. 

The pharmacokinetic/pharmacodynamic of ivermectin varies greatly among species, and, considering the same species, it varies also greatly among individuals, depending mostly on the treated animal’s weight, physiology (in particular fat tissue percentage) and the injection act per se, which is subject to variations (in precision and speed) from an animal to another because animals’ tremors, movements, and because human variability from an injection to another as well. 

Pharmacokinetics variability among the 2 animals of the 2TD treatment (for the reasons mentioned above) might be at the origin of the results observed between TD and 2TD that the reviewer mentioned, with one from both displaying an ivermectin concentration that is as low as the one observed at 14 DAI for the TD treatment (6-7 ng/ml) where variability between animals is low and where this concentration is not associated with significant mortality in mosquito batches fed on these treated pigs. For the same species, inter-individual PK variability is highlighted in table 2 and was already discussed in our manuscript, lines 399-401. This is now mentioned in the results section as well, lines 355 – 356: “For pigs in particular, a great inter-individual host variability in ivermectin plasma concentration can be noticed.” This issue would be solved using more pigs per treatment. However, already published data considered 2 pigs only as well in their protocol (Pasay et al. Parasites Vectors (2019) 12:124 https://doi.org/10.1186/s13071-019-3392-0). 

6.C. Add the time units in the X-axis (days?)

“Days after blood meal” as the time units have been added to the Figure as requested.

7. Line 282-283: From day 14 post-treatment and onwards, there was no significant difference between groups whatever the host species considered (Figure 1).

Should be ¨ Figure 2¨

Response: Thank you very much. This has been corrected in the manuscript at now lines 296-297.

8. Figures 3 and 4: Please, add asterisks to the statically significant differences between control and IVM-treated animals.

Response: Asterisks have been added to the statically significant differences between control and IVM-treated animals as suggested.

9.A. There is no correlation in the mean plasma concentrations of ivermectin (ng/mL) in the treated- sheep, goats and pigs (table II) with the mortality observed in Figure 2 and the number of eggs in Figure 4. Can the authors explain this? 

9.B. For example, in pigs, only the 3TD caused increased mortality on day 14 DPI, but the concentration is as high as 2TD, which did not increase mortality on that day. 

9.C. Besides, 2TD on pigs doubles the concentration in goats at 2 DPI and in sheep at 7 DPI, and both increased mosquitoes’ mortality. A deeper discussion about the lack of correlation in plasma IVM concentrations and the phenotypes observed associated with reduced survival and reproductive fitness is needed. The lack of correlation makes the interpretation of the results very difficult.

9.A and 9.C. Inter-species variability in PK/PD: 

Response: There is evident great discrepancy in plasma levels between species, and not only because the administered doses are different (see Table 2). This has been already reported (Alvinerie et al., 1998; Veterinary Research, 1998, 29: 113-18) and is highly explaine

---

## [Decision Letter · Decision Letter 1]

22 Jul 2024

Impact of blood meals taken on ivermectin-treated livestock on survival and egg production of the malaria vector Anopheles coluzzii under laboratory conditions

PONE-D-23-39281R1

Dear Dr. Pooda,

We’re pleased to inform you that your manuscript has been judged scientifically suitable for publication and will be formally accepted for publication once it meets all outstanding technical requirements.

Kind regards,

Pedro L. Oliveira

Academic Editor

PLOS ONE

Additional Editor Comments (optional):

Reviewers' comments:

Reviewer's Responses to Questions

**Comments to the Author**

1. If the authors have adequately addressed your comments raised in a previous round of review and you feel that this manuscript is now acceptable for publication, you may indicate that here to bypass the “Comments to the Author” section, enter your conflict of interest statement in the “Confidential to Editor” section, and submit your "Accept" recommendation.

Reviewer #1: All comments have been addressed

Reviewer #2: All comments have been addressed

2. Is the manuscript technically sound, and do the data support the conclusions?

Reviewer #1: Yes

Reviewer #2: Yes

3. Has the statistical analysis been performed appropriately and rigorously? 

Reviewer #1: Yes

Reviewer #2: Yes

4. Have the authors made all data underlying the findings in their manuscript fully available?

Reviewer #1: Yes

Reviewer #2: Yes

5. Is the manuscript presented in an intelligible fashion and written in standard English?

Reviewer #1: Yes

Reviewer #2: Yes

6. Review Comments to the Author

Reviewer #1: The manuscript has been significantly improved from its initial version. All of my comments and concerns have been resolved in this new version. I appreciate the authors' efforts in strengthening the discussion about the correlation between plasma levels and the observed phenotypes. After a second revision, I found no further issues to be addressed.

Reviewer #2: The authors have submitted a detailed "answer to reviewers" section. All my comments have been appropriately addressed. All changes in the manuscript are appropriately tracked

7. PLOS authors have the option to publish the peer review history of their article (what does this mean?). If published, this will include your full peer review and any attached files.

Reviewer #1: **Yes: **Marcos Sterkel

Reviewer #2: **Yes: **Carlos Chaccour

---

## [Editor Report · Acceptance letter]

6 Aug 2024

PONE-D-23-39281R1 

PLOS ONE

Dear Dr. Pooda, 

I'm pleased to inform you that your manuscript has been deemed suitable for publication in PLOS ONE. Congratulations! Your manuscript is now being handed over to our production team.

Kind regards, 

on behalf of

Dr. Pedro L. Oliveira 

Academic Editor

PLOS ONE